# Opportunistic Infections and Immune-Related Adverse Events Associated with Administering Immune Checkpoint Inhibitors: A Narrative Review

**DOI:** 10.3390/ph16081119

**Published:** 2023-08-09

**Authors:** Ranferi Ocaña-Guzmán, Diego Osorio-Pérez, Leslie Chavez-Galan

**Affiliations:** 1Laboratory of Integrative Immunology, Instituto Nacional de Enfermedades Respiratorias Ismael Cosio Villegas, Mexico City 14080, Mexico; arocana@iner.gob.mx; 2Department of Medical Oncology, Hospital de la Mujer, Mexico City 11340, Mexico; medicina.diego.osorio@gmail.com

**Keywords:** immune checkpoint, immunotherapy, cancer, IRAE, ICI

## Abstract

Manipulating the immune system by blocking the immune checkpoint receptors is the basis of immunotherapy, a relevant tool in current clinical oncology. The strategy of blocking the immune checkpoints (Immune Checkpoint Inhibitors, ICI) consists of using monoclonal antibodies to inhibit the interaction between ligand and inhibitory receptors from triggering a complete activation of helper and cytotoxic T cells to fight against tumour cells. Immunotherapy has benefited patients with diverse cancers such as stomach, lung, melanoma, and head and neck squamous cell carcinoma, among others. Unfortunately, a growing number of reports have indicated that the ICI treatment also can show a dark side under specific conditions; some of the adverse effects induced by ICI are immunosuppression, opportunistic infections, and organ-specific alterations. This review discusses some immunologic aspects related to these unwanted effects.

## 1. Introduction

In the last decades, immunotherapy (IMT) added Immune Checkpoint Inhibitors (ICI) as one of the most potent tools developed recently to improve traditional oncology treatment. ICI consists of using monoclonal antibodies to inhibit the immune checkpoints (IC)-ligand binding, and, consequently, the activation of the immune cells is conserved [1,2]. Currently, the IMT has been extended to pathologies beyond cancer, but the action mechanisms have been better described in cancer.

The ICs are molecules expressed on the cell surface of diverse immune cells. Some are constitutively expressed, while others are expressed only under specific stimuli. ICs and their ligands modulate the immune response to avoid excessive activation processes and maintain homeostasis; however, in pathologies such as cancer and infections, the expression of ICs is induced to prevent a correct activation of the immune response. Thus, under specific conditions, the IC expression could be harmful [3,4,5].

Monoclonal antibodies against ICs can be divided into murine, chimeric, humanized, and fully human antibodies. The current therapies are focused on targeting mainly 3 ICs: Cytotoxic T Lymphocyte Associated antigen 4 (CTLA-4), Programmed Death-1 (PD-1) and its ligand Programmed Death Ligand-1 (PD-L1), and Lymphocyte Activation Gene-3 (LAG-3) [6,7,8,9]. The US Food and Drug Administration (FDA) has approved using these bio-products as anti-cancer agents that can be used as monotherapy, combined with other ICIs, or as a complement to classic chemotherapy.

Cancer cells express checkpoint molecules or their ligands to escape immune system detection. For instance, CTLA-4 and PD-1 are ICs that limit the T cell activation, allowing tumour growth and cancer progression [3,5]. Reports indicate that cells exposed to an inflammatory microenvironment for an extended period or exposed repeatedly to a specific-antigenic stimulus begin to show a dysfunctional phenotype and overexpression of one or more inhibitory receptors; this phenotype is called “cell exhaustion”, a mechanism where IC plays an essential role in avoiding the immune response [10,11,12,13]. As a result, exhaust cells are less efficient in producing cytokines and proliferating and show reduced effector functions [11,12,14,15].

Indisputably, IMT is one of the major milestones that revolutionized the treatment of multiple solid tumours. Unfortunately, there is also evidence showing unwanted effects among patients treated with immune checkpoint blockers under specific conditions, and it has been termed Immune-Related Adverse Events (IRAE).

The pathophysiology underlying IRAEs is not elucidated yet and appears different for each IMT. Actual information indicates that pathophysiology depends on the role and mechanism of the ICI evolved. For example, some immune checkpoints, such as CTLA-4, inhibit T-Cell activation in the early steps when the immunological synapse begins. In contrast, others, such as PD-1/PD-L1 or LAG-3, inhibit T cells at later stages of the immune response. In this context, reports indicated that knockout mice for CTLA-4 develop lymphoproliferative diseases such as multiorgan lymphocytic infiltration, tissue destruction, myocarditis, and pancreatitis [16]. In contrast to CTLA-4 −/− mice in which most of the CD4+ T cells are non-specifically activated and invade various organs, knockout mice for PD-1 showed autoimmune alteration, including lupus-like proliferative arthritis and myocarditis with high-titter auto-antibodies against cardiac myosin, which indicates that these alterations could be mediated by an antigen-specific autoimmune response [17,18].

The molecular mechanism under these adverse events is not known. Postow and collaborators [19] postulate four hypotheses about the trigger mechanism for IRAEs after ICI treatment:ICIs increase T cell activation against antigens expressed in tumour cells and healthy tissue.ICIs increase levels of pre-existent auto-antibodies.ICIs increase the level of inflammatory cytokines.Anti-CTLA-4 antibodies cause complement-mediated inflammation, impairing normal tissue that expresses CTLA-4.

However, more mechanisms may exist associated with the development of IRAEs, or even a combination of these mechanisms could be activated simultaneously. This review aims to explore recent clinical evidence of the most frequent IRAEs to try to understand immunological mechanisms and the associated pathologies.

## 2. Immune Checkpoints: How Do They Induce Inhibition of Activated T Cells?

Knowing the basic molecular mechanisms of T cell activation is imperative to comprehend how the ICs block this activation and, consequently, how the ICI works. The T cell receptor (TCR) is an antigen-specific receptor expressed on T cells. It is generated through a gene recombination process during early T-cell development. TCR recognizes the antigen, which is presented by the molecules’ histocompatibility complex (MHC); posterior to the interaction TCR–antigen–MHC, several kinases such as Lck and ZAP-70 are recruited downstream of the TCR, and the domains’ ITAM (immunoreceptor tyrosine-based activation motif) of the CD3 chains, which are coupled to the TCR, are phosphorylated [20,21].

The second signal necessary to induce an adequate T cell activation is called the “co-stimulation”, which is given by the interaction CD28/CD80 (B7-1) and CD86 (B7-2). This interaction is requested to expand signals to molecules such as LAT, PLC-γ, Grb2, and SOS, among others, to activate the Ras and MAP-Kinase signalling to favour the Ca^2+^ flux. Thus, activated T-cells start functions such as proliferation, cytokines production, receptors expression, and production of molecules associated with their immune function, according to the T cell subpopulation (Figure 1A) [22].

Inhibitory receptors induce tolerance or suppression through several mechanisms. One of the most well-described is mediated by the receptorPD-1 and their ligands’ Programmed Death Ligand 1 and 2 (PD-L1 and PD-L2, respectively). PD-1 is an inhibitory receptor, a member of the immunoglobulin superfamily, and initially associated with the regulation of cell death; structurally, PD-1 has an ITIM domain [23]. ITIM domains, the opposite of ITAM domains, are associated with tyrosine phosphatases such as SHP-1 (Src homology 2 domain-containing protein tyrosine phosphatase 1), enzymes that neutralize the effect of ITAM domains. Consequently, the activation process of T cells is affected [24] (Figure 1B).

Another inhibitor receptor is CTLA-4, structurally similar to CD28, a ligand to CD80 or CD86 [25]. Furthermore, evidence supports that expression of CTLA-4 obstructs T-cell activation by reducing co-stimulation [26]. Thus, overexpression of this receptor on T cells causes a down-regulation of the signal because CTLA-4 competes for CD80 and CD86; consequently, the co-stimulation signal is inhibited (Figure 2A).

LAG-3 is a particular IC, discovered in 1990 by Triebel and colleagues [27], and is structurally related to CD4 protein. However, little is known about its molecular function. LAG-3 has been proposed to bind to MHCII with higher affinity than CD4 and inhibit T cell activation by interfering with the association of CD4 with MHCII; in this regard, the relative affinity was established using immunoglobulin fusion proteins for CD4 and LAG-3 [28] (Figure 2B).

The T cell immunoglobulin and mucin-domain containing-3 (TIM-3) is another IC. It was initially known as a membrane-specific marker for Th1 and Tc1 lymphocytes. However, TIM-3 expression was soon identified in other cell types as monocytes, macrophages, dendritic cells, and NK cells [29,30,31]. TIM-3 can regulate T-cell activation through the interference of the TCR pathway [32]. Lee et al. demonstrate that TIM-3 interaction with Galectin-9 (Gal-9) induces intracellular tail phosphorylation TIM-3 and subsequent recruitment of Lck kinase, a relevant component in the TCR pathway [33].

Thus, ICs are a heterogeneous group of protein receptors that regulate the activation of immune cells at different levels to induce peripherical tolerance and avoid an exacerbated response that can cause tissue damage.

## 3. IMT and Cancer: Successful Relation

Compared to chemotherapy drugs such as cisplatin and carboplatin, IMT has shown high specificity and lower toxicity. The IC are some of the most important therapeutic targets identified in recent years. IMT has been focused mainly on cancer since the first approval of ipilimumab in 2011 [34]. The more common strategy is using a high-affinity monoclonal antibody to block the inhibitory receptors. In this way, ICI blockade induces a complete reactivation of the host immune system that permits the attack of tumour cells [35]. Table 1 summarises the current IMT used.

In 2022, Cercek et al. published a prospective study on Dostarlismab, which is a PD-1 inhibitor. All 12 patients who participated in the study, showed remission of rectal adenocarcinoma; those patients received Dostarlismab every 3 weeks for 6 months [36]. This result was added to a growing list of reports with outstanding results in diverse cancer types [37,38].

The IMT benefit in the field of cancer is unquestionable; the World Cancer Report 2020 indicated that IMT-use is the most relevant addition to anti-cancer therapy. Unfortunately, ICI therapy does not work for all patients. A 2020 study estimates that 43.6% of US cancer patients are eligible for ICI therapy in the US, with 12.5% showing beneficial responses to ICIs. Chemotherapy and radiotherapy are still the main treatment for cancers, but the IC blockade could soon be the first line of treatment for many solid and blood tumours. Some of the treatment schemes used currently have target PD-1, CTLA-4, and PD-L1 (Table 1); however, in oncology, immunotherapy research is a field in constant growth, and the IMT repertory is highly likely to increase in a short time.
pharmaceuticals-16-01119-t001_Table 1Table 1Authorized immunotherapy in cancer treatment.DrugTargetTypeRefs.NivolumabPD-1Human IgG4, anti-PD-1[39,40,41]PembrolizumabPD-1Humanized IgG4, anti-PD-1[37,42,43,44]AtezolizumabPD-L1Humanized IgG1, anti-PD-L1[45,46,47]CemiplimabPD-1Human IgG4, anti-PD-1[48,49]IpilimumabCTLA-4Human IgG1, anti-CTLA-4[8,50,51]AvelumabPD-L1Human IgG1 anti-PD-L1[52,53,54,55]DurvalumabPD-L1Human IgG1κ anti-PD-L1[7,56,57]DostarlimabPD-1Humanized IgG4 anti-PD-1[58,59,60]TremelimumabCTLA-4Human IgG2 anti-CTLA-4[56]RelatlimabLAG-3Human IgG4 anti-LAG-3[9,61]


## 4. Infectious Diseases Associated with the Use of IMT

In addition to the IRAEs reported using ICI, the appearance of opportunistic infections is associated with immunosuppressive treatments, which paradoxically are used to prevent IRAEs. Although many opportunistic infections can be treated with classic antibiotics, some can present severe forms with fatal outcomes [62,63].

Below are discussed some of the leading infectious diseases associated with IMT in patients with cancer, which are the principal goals for IMT use.

### 4.1. Tuberculosis (TB)

TB is an infectious disease caused by the bacilli *Mycobacterium tuberculosis* (M.tb). The immune response develops a cellular structure called granuloma to contain the M.tb infection. TNF (Tumour Necrosis Factor) is an inflammatory cytokine necessary to maintain and form granuloma [64,65]. TNF was one of the first molecules selected as a target in the treatment of rheumatoid arthritis (RA) to limit chronic inflammation; infliximab is an anti-human monoclonal antibody (chimeric mouse–human) that inhibits soluble TNF in both the monomeric and trimeric form, and etanercept is a fully human recombinant molecule consisting of two subunits of the TNF receptor 2 to block the trimeric form of TNF [66]. From 1998 to 2001, 70 TB cases were associated with using infliximab; from these, 12 patients died [67].

Now in the era of ICIs, reports indicate a substantial risk of latent TB reactivation associated with using blockers of the PD-1 pathway. Using a mouse model knockout for PD-1, E. Lázár-Molnár et al. reported that the loss of the PD-1 pathway favours an excessive inflammatory state, increasing the necrotic damage and reducing the infiltration of T and B cells, which affects the capacity to control M. tb proliferation [68].

A growing number of reports show a reactivation of pulmonary TB in humans using IMT, for instance, after using PD-1 inhibitors such as Pembrolizumab and Nivolumab [69,70,71]. The frequency of TB reactivation is higher in patients with haematological malignancies compared to solid tumours [72,73]. A meta-analysis by K. Liu et al. indicates that patients treated with PD-1/PD-L1 blockers had a 35 times higher probability of reactivating TB than the general population, and the mortality was extremely high because 30% of these patients that experienced reactivated TB died [73].

Even patients with uncommon cancer, such as nasopharyngeal carcinoma and Merkel cell carcinoma, developed TB following PD-1 blockade therapy; authors described that pembrolizumab disturbs the T cell response because, although they observed specific Th1 lymphocytes CD4+, others such as Th17, CD8+, and FoxP3+ T-cells did not show an increase over time [69].

The limited evaluations of specific T cells to cancer cells and M.tb suggest an imperative need for animal models at a basic level and the clinical caution to test cancer patients for latent TB before starting ICI treatment, especially in countries with a high incidence of TB.

### 4.2. Aspergillus fumigatus Infection

*Aspergillus fumigatius* is a fungus, causative of several diseases, mainly in the respiratory airways due to the high presence of spores in the airborne particulate matter [74]. As a result of the immune status of the host, *Aspergillus fumigatius* spores can lead to a broad spectrum of diseases, including invasive aspergillosis, frequently affecting patients with chronic obstructive pulmonary disease. Other forms are chronic pulmonary aspergillosis and allergic bronchopulmonary aspergillosis, which show a high incidence among asthma and cystic fibrosis patients [75]. Diverse reports have demonstrated isolated cases of patients with *Aspergillus fumigatus* infection after IMT use.

A 62-year-old man with diabetes mellitus and metastatic renal cell carcinoma was initially treated with monoclonal Nivolumab and Ipilimumab to block PD-1 and CTLA-4, respectively [76]. Posterior to the finish of the scheme, this patient continued using rituximab for the carcinoma treatment [76]. In a separate case from the Brooklyn Hospital Center (New York), a 63-year-old man completed chemotherapy with paclitaxel and carboplatin to treat non-small-cell lung cancer (NSCLC). After three months, four cycles of Durvalumab began to block PD-L1; however, the patient showed difficulty of breath, and *Aspergillus fumigatus* was identified in the pleural fluid culture [77].

Another reported case is a 68-year-old man treated for NSCLC with chemoradiotherapy and subsequently with durvalumab [78]. After the second dose, bacterial pneumonia with fever and IRAE were considered, and *Aspergillus fumigatus* was isolated from bronchoscopy-obtained samples [78]. This patient did not report comorbidities associated with infection susceptibility such as diabetes mellitus; it is important to note this because data suggest that opportunistic infections in patients with IM could be related to metabolic disorders [77].

The exact mechanism by which opportunistic infections occur in patients with IMT is poorly understood. Still, it is possible that the co-infections could result from many immune alterations during ICI administration, such as a hyper-inflammatory state. Usually, to control this paradoxical response, corticosteroids are used to reduce hyper-inflammation. However, this induced immunosuppression also induces another undesirable effect.

### 4.3. Pneumocystis jirovecii

Pneumonia caused by the fungus *Pneumocystis jirovecii* is a common opportunistic infection affecting immunosuppressed patients [79]. Currently, there are reports about this infection in patients treated with IMT.

M. Schwarz et al. reported two lethal cases of infection by *Pneumocystis jirovecii* in patients treated with PD-L1 blockers [80]. The first was a man, 79 years old, diagnosed with bilateral NSCLC, posterior to an unsuccessful scheme of six cycles of chemotherapy with carboplatin and gemcitabine, followed by radiotherapy; nivolumab was initiated, but it was stopped twice because of recurrent respiratory infections. A thoracic computerized tomographic (CT) scan showed that the patient had reticular and nodular thickening at the four nivolumab cycles. Nivolumab was retired, and immunosuppressive treatment with corticosteroids was initiated. However, the patient presented severe dyspnoea, dry cough, hypotension, tachycardia, and fever after four weeks, and, using quantitative PCR, the presence of *Pneumocystis jirovecii* was confirmed. The patient was deceased 2 weeks later due to respiratory failure [80].

In the second case, a man of 53 years old was diagnosed with stage IIIA NSCLC. The patient was treated with cisplatin and vinorelbine, followed by a right upper lobe resection [80]. Nivolumab and radiotherapy were used to treat residual mediastinal tumour persistence. By the third cycle of nivolumab, the patient developed dyspnoea and fever, and the CT scan showed evidence of pneumonitis. Nivolumab was suspended from initiating corticosteroid administration, despite an initial improvement; after a month, the patient deteriorated dramatically, and his bronchial alveolar lavage was positive for *Pneumocystis jirovecii* infection and cytomegalovirus positivity [80].

There are reports of complications in younger people than in the previous study. A report of 2020 at the Children’s Hospital of Philadelphia describes the case of an 18-year-old woman diagnosed with primary mediastinal B-cell lymphoma (PMBCL) and a previous history of pulmonary embolism. Initially, she was treated with six cycles of rituximab, etoposide, prednisone, vincristine, cyclophosphamide, and hydroxydaunorubicin with a refractory result. Then, the patient began treatment for 8 months with pembrolizumab, and at 11 doses, she started to have respiratory symptoms of exertional shortness of breath, chest tightness, and pleuritic chest pain. *Pneumocystis jirovecii* was detected in the bronchoalveolar lavage fluid, and she was treated with 21 days of high-dose trimethoprim/sulfamethoxazole and steroids. After 7 days, she left the hospital, and two weeks later, she improved [81].

Thus, *Pneumocystis jirovecii* infection is a severe condition that occurs in immunocompromised patients from IMT use, and it often leads to fulminant respiratory failure. At present, it is not clear which immune mechanism is affected to induce IRAEs using IC blockers. We suggest that the evaluation of the frequency of T cell subpopulations should be included during the follow-up of patients under IMT schemes. This information could provide relevant knowledge to identify the main cell subpopulation affected when a specific IC is blocked, even though it is not clear whether the cell subpopulations are involved in a specific way with the blocked molecule or if the IRAEs are a consequence of a general immune alteration.

## 5. Organ Damage by the Use of IMT, Even in the Absence of Pathogens

An extensive review by V. Vasconcellos et al. [82] indicates that carboplatin has an elevated risk of myelosuppression and neurotoxicity; meanwhile, cisplatin is related to a high rate of nausea and vomiting, nephrotoxicity, and ototoxicity.

Patients who are treated with ICI have variable risks of developing toxicities. Side effects of immune therapy vary from mild manifestations such as fatigue and diarrhoea to more serious complications such as hepatic and renal toxicities or even rheumatological disorders (Table 2).

The toxicity profile differs between monoclonal antibodies to block CTLA-4, PD-1/PD-L1, and LAG-3. ICIs combined treatments or in combination with classic chemotherapy may present increased toxicity because of the sum of the side effects of each antibody, increasing complications; although in many cases, the effectiveness of these ICIs has also been demonstrated [41,42,84,101,102,103]. The highly frequent IRAEs developed by cancer patients treated with IMT, which are produced in the absence of pathogens, are shown in Figure 3.

### 5.1. Adverse Hepatic Events

Hepatitis is one of the most common IRAE-associated diseases with the uses of anti-CTLA-4, anti-PD-1, and their ligands. Alterations include elevation of aspartate aminotransferase, alanine aminotransferase, and, in most cases, raised bilirubin and fever [104]. The incidence of hepatitis in patients treated with anti-PD-1 ICI is approximately 5%, but this rises to 30% in patients treated with a combination of CTLA-4 and PD-1 blockers [105].

These adverse effects can also be presented with the reactivation of latent infections. T.H. Kim documented a clinical case of a 60-year-old patient with advanced lung cancer stage IV who had been treated with nivolumab for 15 months [84]. Metastasis was discarded, but PCR confirmed the TB diagnostic. However, the patient showed altered test liver function by combining the anti-TB and nivolumab treatments. This report indicates the relevance of focusing on liver function changes in combination treatment with nivolumab and anti-TB [43].

K. Adaman and collaborators established the B6/lpr mice as a model to replicate the adverse effects of IC treatment [106]. These mice develop hepatitis, characterized by mononuclear cell infiltration to the periportal and pericentral hepatic veins, after receiving anti-PD-1 and anti-CTLA-4 [106]. Using this model, T CD4+ lymphocytes are the main cell population present in liver infiltration, suggesting that T cell infiltration could be one of the causes of hepatitis in humans after ICI treatment.

### 5.2. Pulmonary Adverse Events

The lung is one of the organs with elevated frequency of IRAEs among patients treated with IC. Interstitial lung disease (ILD) is a disease that affects the parenchyma or alveolar regions. A report indicated that 8 of 111 patients with recurrent or advanced non-small-cell lung cancer (NSCLC), followed in a clinical trial stage II, showed ILD after three doses of nivolumab, and 7 of these cases were considered as adverse events related to nivolumab treatment [6].

Similar results in a cohort of 119 NCSCLC-stage IV patients were reported [83]. Among these patients, seven displayed signs of lung obstruction caused by tumour-mediated compression at the initiation of nivolumab treatment, three developed acute ILD within 10 days of treatment with nivolumab, and two of them died [83]. Despite the fact that the pneumonitis following anti-PD1 treatment were been reported as a complication [107], these reports indicate that severe and acute forms of ILD can develop in a very short period after anti-PD1 therapy.

Moreover, the model of B6/lpr mice also showed that there is a high cell infiltration in the lung after anti-PD-1/anti-CTLA-4 [106]. The tumour size and cell infiltration have an inverse correlation, suggesting that, although the over-activation of immune cells is the cause, tumour reduction is also participating in the development of IRAE.

Other lung alterations less frequent than Nivolumab-induced asthma have been documented; in 2017, a 50-year-old man with no prior medical history was diagnosed with stage IV lung adenocarcinoma [86]. Despite his having no history of familiar asthma or atopy, he presented at the hospital with cough and wheezing, which worsened at night and in the early morning nine months after start treatment with anti-PD-1 nivolumab. Clinical parameters such as high levels of IgE and eosinophilia were identified, in addition to altered spirometry values. The indicated treatment was daily inhalation of fluticasone propionate/formoterol fumarate dehydrate. His symptoms were rapidly controlled in a few days [86].

The mechanism evolved in this rare phenomenon is not well identified, but reports exist that suggest that the PD-1 axis can be evolved in airway alterations. Asthma is a chronic airway inflammation in which airflow obstruction and airway hyperresponsiveness are present. In a murine model of asthma, Akbari O. et al. report that the mouse PD-L2 −/− shows increased airway hyperresponsiveness and lung inflammation compared with wild-type mice [108].

### 5.3. Dermatological Toxicity Events

One of the most common side effects of IC treatment is dermatological reactions such as rash or pruritus, which have been reported in high frequency (±62%) with grades 1 and 2 in severity among patients treated with ipilimumab or nivolumab [8,109]. A higher frequency of skin alterations is associated with the combination of anti-CTLA-4 and anti-PD-1 treatment [41,102]. Because the skin is an organ enriched in lymphocytes, it is highly affected, and cutaneous IRAEs of the skin occur quickly and with high frequency.

In several studies, it appears that toxicity in the skin differs between IC targets; for example, in patients treated with anti-PD-1 antibodies, vitiligo is seen more frequently, and, in contrast, patients treated with anti-CTLA-4 antibodies are seen with rash and pruritus more often [39,110,111,112]. It is necessary to mention that skin IRAE incidences increase in patients with a clinical history of skin-associated pre-existing autoimmunity, such as psoriasis and lupus [17,113,114].

Eczematous rash (ER) is one of the most frequent adverse reactions. Er is characterized by erythematous macules and papules with pruritic content in the larger papules. This IRAE occurs in up to 68% of patients on anti-CTLA-4 and up to 20% on anti-PD-1/PD-L1 treatment [115]. The onset is usually between 3 and 6 weeks after ICI administration [112,116]. The psoriasis-like rash is another complication that can be present during treatment with ICIs. However, the most common state is the exacerbation of pre-existing psoriasis, and de novo psoriasis is rare [113,114]. The most frequent is the Plaque-type psoriasis. Other less common types, such as pustular, scalp, and inverse psoriasis, can also occur [114]. A vitiligo-like rash is a common form of IRAE developed mainly in patients with melanoma treated with anti-PD-1/anti-PD-L1. In this case, progressive skin depigmentation can appear in approximately 25% [110,117]

The pathophysiological and immunological mechanisms of ICI-induced cutaneous IRAEs are not elucidated yet, but some aspects have been identified; the main cases involve the CD4+, CD8+ T cell overactivation and increased stimulation of B cells, innate cells, and pro-inflammatory cytokines production [88,111,118]. A previous evaluation of the T cell phenotype, by flow cytometry, in patients with stage IV melanoma or renal cell cancer treated with an anti-CTLA-4 reported an increased frequency of T cell subtypes such as CD4+, CD8+, CD4+CD25+, and CD4+CD25- that co-express the activation marker HLA-DR [119]. Additionally, results indicate that regulatory T cells CD4+CD25+FoxP3+ are not affected by anti-CTLA-4 treatment, and this is in conjunction with histological data obtained from several studies that report perivascular lymphocytic infiltrate extending deep into the dermis and epidermis [97,99]. Evidence shows that blocking PD-1 and CTLA-4 promotes TH-1 and TH-17 activity. TH-17 cytokines such as IL-17 and IL-22 promote neutrophil recruitment and keratinocyte proliferation [120].

A disadvantage of the study of dermatological toxicities associated with IC treatment is that the mouse model that K. Adam and collaborators established did not replicate skin alterations and inflammation states shown in human patients [106].

## 6. Conclusions

Immunotherapy represents the most relevant addition to the clinical oncology field. The advantage of ICI antibodies has been proven in multiple basic and clinical assays; diverse evidence supports the efficacy of monoclonal antibodies to block immune checkpoint receptors. In this way, the effectiveness and advantages of immune therapy are not discussed; however, the side effects should be considered a priority aspect that should be addressed with basic and clinical trials.

In addition to ICI, there will soon exist other clinical approaches to treat cancer. A relevant approach to the problem cited in this review is the use of non-peptidic compounds that can inhibit the receptor-ligand interaction of immune checkpoints. In this regard, two examples of these small molecule inhibitors are the molecules termed BMS-8 and BMS-202 (Bristol-Myers Squibb-8 and -202, respectively), which is a derivative of (2-methyl-3-biphenylyl) methanol [121]. BMS-8 and BMS-202 can dissociate the PD-1/PD-L1 interaction by binding to the surface of “hot spots of PD-L1”. One of the most valuable characteristics of these immunomodulatory molecules is the reduced toxicity, which confers an excellent advantage compared with monoclonal antibodies [121]. 

Currently, a growing body of evidence suggests alternatives to decrease the toxicity effect of the ICI—for instance, nanoparticles and nanoengineering technology that have been proven in preclinical studies. One of the most recent examples is the COVID-19 vaccines developed by BioNTec, which use lipid nanoparticles to deliver antigen-coding messenger RNAs (mRNAs) [122,123]. Under the cancer context, nanoparticles with tumour-associated antigens (TAAs) have been one of the strategies proved and approved by the FDA’s DC-based sipuleucel-T (Provenge^®^) [124]. Nanocarriers can improve ICI efficacy by modulating their pharmacological characteristics. For example, it has been demonstrated that nanoparticles based on polylactide-co-glycolide (PLGA) polymer can increase the accumulation of anti-PD-1 antibodies in the spleen, with a subsequent uptake by dendritic cells, which induces maturation and activation of DCs [125].

Another example is the use of nanoparticles to reduce the expression of ICs. In a study using a mouse model, lipid bilayer-coated mesoporous silica nanoparticles were filled with a small-molecule inhibitor of the signalling hub kinase, glycogen synthase kinase-3 (GSK3), which was created to interfere with the PD-L1/PD-1 axis by suppressing PD-1 expression [126]. 

Finally, the use of thermal immuno-nanomedicines is another option to incorporate immunotherapies. In this strategy, thermal immuno-nanomedicines can induce local hyperthermia, promoting cancer cell killing with tumour shrinkage and beneficial changes in the tumour microenvironment [127].

At present, the irruption of monoclonal antibodies for cancer treatment represents the newest and most potent tool incorporated into cancer treatment. This fact highlights the importance of effectively identifying and treating the risks of using ICI.

Despite the advantages of these clinical strategies, we must recognize a considerable delay in immunomodulatory compounds and nanocarriers compared to the development of monoclonal antibodies. It is possible that, in the future, these technologies can be applied together or as a complement to current anti-cancer treatments, which represents an indisputable benefit.

In this sense, there is an urgent need to identify and even prevent the side effects that monoclonal antibodies have by blocking an inhibitory pathway, both in normal and cancer cells; they must be evaluated using in vitro and in vivo models. The increasing number of reports of SARIs following ICI treatment indicates that there are adverse reactions that have not been fully identified as risk factors in these biologics’ development and clinical trials.

In addition to the hypotheses about the mechanism underlying the toxicity associated with using ICI, it is necessary to continue exploring other mechanisms and the possible toxic effect resulting from the synergy between ICI and other therapeutic agents.

To treat organo-specific reactions, new guidelines indicating the most frequent adverse reactions and the drug and recommended dose must be indicated. However, additional analysis must be performed before ICI administration in the case of opportunistic infections. Considering infectious diseases’ prevalence in each demographic zone, analysis, such as PCR or antigen test, must be indicated to all patients who potentially can be treated with ICI therapy.

A most extensive evaluation of the immune-exhausted state of patients is required before starting ICI treatment to identify Immune checkpoint expression and possible biomarkers that can help to predict the adverse effect of immunotherapy.

## Figures and Tables

**Figure 1 pharmaceuticals-16-01119-f001:**
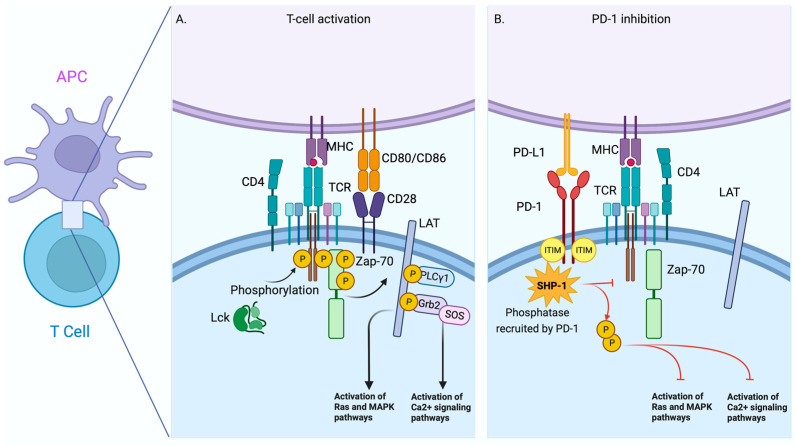
Comparative scheme between T-Cell activation by TCR and T-Cell Inhibition mediated by PD-1. Schematic view of the early signalling events of the interaction between T cells and antigen presenting cells (APCs) during T cell activation (**A**). The interaction of PD-1 and PD-L1 causes the recruitment and activation of phosphatases such as SHIP-1 that disrupt the activation pathway in T cells (**B**). Red line = Blocking signal. The figure was created in BioRender.

**Figure 2 pharmaceuticals-16-01119-f002:**
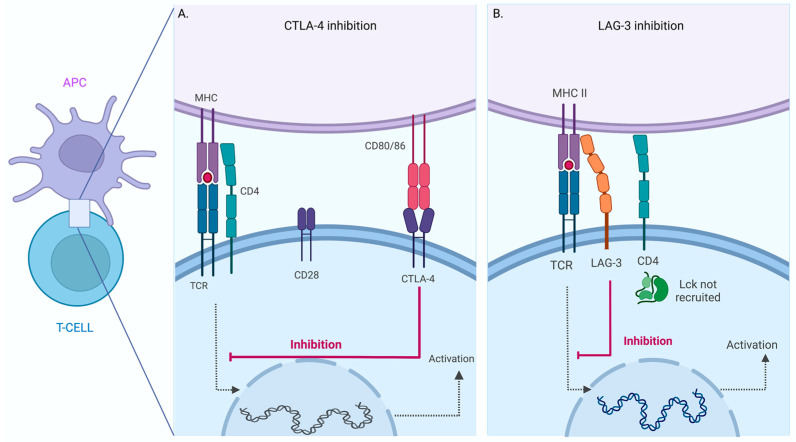
Inhibition of T -Cell activation mediated by CTLA-4 and LAG-3. Schematic view of T Cell-Antigen Presenting Cell (APC) interaction. Inhibition of CTLA-4 (**A**) and LAG-3 (**B**) mediated T cell activation. Schematic view of the T cell-antigen presenting cell (APC) interaction. The activation pathway is triggered when a specific TCR recognizes the MHC–peptide complex (dotted line). The inhibition mechanism can occur in the same cell when CTLA-4 interacts with CD80/86 or LAG-3 binds MHC. The binding of these inhibitory receptors results in blocking the activation process in the T cell (red line). The figure was created in BioRender.

**Figure 3 pharmaceuticals-16-01119-f003:**
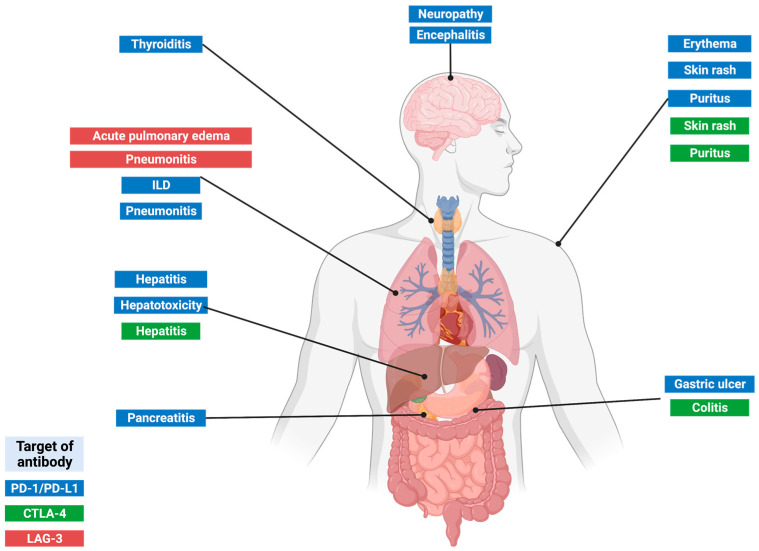
Main IRAEs associated with the ICI use.

**Table 2 pharmaceuticals-16-01119-t002:** IRAE reported by specific ICI.

Drug	Target	Organ Damage Induced by ICI	Refs.
Nivolumab	PD-1	Hepatotoxicity, ILD, rash, pruritus, asthma, pancreatitis.	[6,83,84,85,86,87]
Pembrolizumab	PD-1	Meningitis, thrombosis, gastric ulcer, thrombocytopenia, adrenal insufficiency, cutaneous erythema.	[88,89,90,91,92,93]
Atezolizumab	PD-L1	Skin rash, pneumonitis, hepatotoxicity, peripheral neuropathy, encephalitis.	[94,95,96]
Ipilimumab	CTLA-4	Hepatitis, rash, pruritus, colitis, thyroiditis.	[97,98,99]
Avelumab	PD-L1	Thyroiditis, rash and pruritus, hepatitis.	[54,100]
Durvalumab	PD-L1	Lung damage	[78]
Dostarlimab	PD-1	Fatigue, diarrhoea and nausea, neutropenia	[59,60]
Relatlimab	LAG-3	Increased levels of lipase, acute pulmonary oedema, and pneumonitis	[9]

## Data Availability

Not applicable.

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
