# Peer review of "Opportunistic Infections and Immune-Related Adverse Events Associated with Administering Immune Checkpoint Inhibitors: A Narrative Review"

_pharmaceuticals, 2023, doi:10.3390/ph16081119_

Round 1

Reviewer 1 Report

This review discusses some immunologic aspects related to these unwanted effects, which is helpful to further improve the application of ICI in clinic. Before acceptance, the following points should be addressed.

1. Some latest references on immunotherapy should be cited, such as Nature Reviews Clinical Oncology, 2023, 20 (2): 116-134; Int J Biol Sci. 2023; 19(8): 2428–2442.

 2. Some small molecular inhibitors, like BMS202, possesses several advantages compared to monoclonal antibodies, which may also one way to reduce the toxicity of ICI therapy. And the following references should be cited. Advanced Science, 2023, 10 (3): 2203788; Oncotarget, 2016, 7(21): 30323–30335.

3. Using nanocarrier is also an effective way to decrease the side effects of monoclonal antibodies. Thus, the author could emphasize this point in last section.

Author Response

Dear Reviewer, we appreciate your comments, all of them improved our manuscript.

  1.  Some latest references on immunotherapy should be cited, such as Nature Reviews Clinical Oncology, 2023, 20 (2): 116-134; Int J Biol Sci. 2023; 19(8): 2428–2442.

R: Thanks for your recommendation. Both references have been cited in the new version.

  1.  Some small molecular inhibitors, like BMS202, possesses several advantages compared to monoclonal antibodies, which may also one way to reduce the toxicity of ICI therapy. And the following references should be cited. Advanced Science, 2023, 10 (3): 2203788; Oncotarget, 2016, 7(21): 30323–30335.

R: Thank you for your observation. We have included information regarding points 2 and 3 in the paper's final section, which has enriched the review by providing a broader overview of new clinical tools for cancer treatment.

  1. Using nanocarrier is also an effective way to decrease the side effects of monoclonal antibodies. Thus, the author could emphasize this point in last section.

R: Thanks for your recommendation.  We have included information  in the paper's final section

Reviewer 2 Report

  1. The text mentions "SHP-1 (The Src homology 2 containing tyrosine phosphatase-1)" as an enzyme that neutralizes ITAM domains. The correct full form for SHP-1 is "Src homology 2 domain-containing protein tyrosine phosphatase 1."
  2. The paragraph discussing the association of LAG-3 with MHCII implies that LAG-3 binds to MHCII with higher affinity than CD4. However, the exact affinity comparison between LAG-3 and CD4 for MHCII is not well-established.
  3. The paragraph that introduces the concept of immune checkpoint inhibitors (ICIs) should provide a clear definition or explanation of what ICIs are before diving into specific examples. It may be helpful to provide a brief introduction to ICIs and their role in cancer therapy.
  4. In the section discussing "IMT and cancer: Successful relation," the text mentions a prospective study by Cercek et al. in 2022 that showed a 100% remission rate of rectal adenocarcinoma with Dostarlismab (a PD-1 inhibitor). However, a 100% remission rate in a cancer study is highly unusual and may be an exaggeration or misinterpretation of the study's findings (Cercek states that longer follow-up is needed to assess the duration of response).
  5. The text mentions that approximately 20% of patients have access to ICI therapy, but it doesn't provide any context (only the world cancer report) for this statistic, making it unclear where this information comes from.

There are some punctuation and spacing errors throughout the text, including missing commas, misplaced semicolon, and inconsistent spacing after punctuation marks.

The text lacks a clear and well-structured introduction that defines the topic, presents the research question or objective, and provides an overview of the paper's content. The introduction should also provide context and relevance to the scientific community.

Some sections of the paper jump abruptly between topics, making the flow and coherence unclear. The paper needs to be reorganized and properly structured to present the information in a logical manner.

Author Response

Dear reviewer, we appreciate your comments, all of them improved our manuscript.

  1. The text mentions "SHP-1 (The Src homology 2 containing tyrosine phosphatase-1)" as an enzyme that neutralizes ITAM domains. The correct full form for SHP-1 is "Src homology 2 domain-containing protein tyrosine phosphatase 1."

R: Thank you for the correction. The complete name of SHP-1 was corrected.

  1. The paragraph discussing the association of LAG-3 with MHCII implies that LAG-3 binds to MHCII with higher affinity than CD4. However, the exact affinity comparison between LAG-3 and CD4 for MHCII is not well-established.

R: Thanks for your observation. The affinity of CD4 and CTLA-4 for MHC-II has been evaluated using fusion proteins and comparing their binding to MHC-II expressed in b lymphocytes and fibroblasts. The CD223 (LAG-3) fusion protein showed a higher relative affinity to MHC-II compared to the CD4 fusion protein.

We consider that it could be better not to mention the complete mechanism in the main text to avoid diverting attention from the main topic. However, it is mentioned in lines 122-123, and the new reference #28 was added.

Ref: Workman CJ, Rice DS, Dugger KJ, Kurschner C, Vignali DA. Phenotypic analysis of the murine CD4-related glycoprotein, CD223 (LAG-3). Eur J Immunol. 2002 Aug;32(8):2255-63. doi: 10.1002/1521-4141(200208)32:8<2255::AID-IMMU2255>3.0.CO;2-A. PMID: 12209638.

  1. The paragraph that introduces the concept of immune checkpoint inhibitors (ICIs) should provide a clear definition or explanation of what ICIs are before diving into specific examples. It may be helpful to provide a brief introduction to ICIs and their role in cancer therapy.

R: Thank you for your kind observation. A brief description of immune checkpoint inhibitors can be found in the article at the beginning of the introduction (lines 27-32), and later we have extensively described the characteristics of ICIs.

  1. In the section discussing "IMT and cancer: Successful relation," the text mentions a prospective study by Cercek et al. in 2022 that showed a 100% remission rate of rectal adenocarcinoma with Dostarlismab (a PD-1 inhibitor). However, a 100% remission rate in a cancer study is highly unusual and may be an exaggeration or misinterpretation of the study's findings (Cercek states that longer follow-up is needed to assess the duration of response).

R: Thank you for your kind observation. We have changed the expression "100%" to give the reader a conservative perspective of the result and to leave the findings of that study to their interpretation.

  1. The text mentions that approximately 20% of patients have access to ICI therapy, but it doesn't provide any context (only the world cancer report) for this statistic, making it unclear where this information comes from.

R: Thank you for your kind observation. We have reviewed the WHO world cancer report again, and unfortunately, it does not state how the 20% estimate was derived. We have modified the text and included a precise study, which estimates the percentage of patients who are eligible to receive ICI therapy.

Ref: Haslam A, Gill J, Prasad V. Estimation of the Percentage of US Patients With Cancer Who Are Eligible for Immune Checkpoint Inhibitor Drugs. JAMA Netw Open. 2020 Mar 2;3(3):e200423. doi: 10.1001/jamanetworkopen.2020.0423. PMID: 32150268; PMCID: PMC7063495.

Finally, the language was revised to provide a better structure to the manuscript.

Reviewer 3 Report

The authors have summarized the opportunistic infections and immune-related adverse events associated with the administration of immune checkpoint inhibitors. Generally, this review is interesting but there are still some points to be addressed.

1The frame of the abstract needs to be reorganized.

2) More latest references are suggested to include.

3) Not enough contents in section "5.3 Dermatological toxicity events".

4Figure 2 is mainly talking about LAG-3 rather than TIM-3.

5) The quality of figures needs to be improved. 

6) There are some spelling mistakes such as “whit" or grammar errors, like "CTLA-4 or PD-1 are".

7) Some format needs to be unified. For instance, Line 71-80; and Line 93.

Language should be further polished.

Author Response

Dear reviewer, we appreciate your comments; they all improved our manuscript.

1) The frame of the abstract needs to be reorganized.

R: The abstract has been reorganized in the new version of the manuscript.

2) More latest references are suggested to include.

R: Thank you for your kind observation. We have included new and more recent references, greatly enriching the text.

3) Not enough contents in section "5.3 Dermatological toxicity events".

R: Thank you for your kind observation. The mentioned section has been expanded to provide a broader overview to readers.

4) Figure 2 is mainly talking about LAG-3 rather than TIM-3.

R: Thank you for the indication. The reference to Figure 2 was incorrect and now appears at the end of the correct paragraph where we talk about LAG-3.

5) The quality of figures needs to be improved. 

R: We included figures with the maximum quality provided by BioRender.

6) There are some spelling mistakes such as “whit" or grammar errors, like "CTLA-4 or PD-1 are".

R: We apologize for those mistakes; the incorrect word has been replaced.

7) Some format needs to be unified. For instance, Line 71-80; and Line 93.

R: Thank you for your kind observation. The indicated lines show a different format because we want to list 4 proposed mechanisms for the immunopathogenesis of IRAEs. We believe it could be clear for the reader to highlight these aspects with this format change.

Finally, the manuscript was submitted for a language revision to avoid grammatical errors. 

Round 2

Reviewer 2 Report

Figures could benefit from discrete categorization, denoted as A and B.

Within Figure 3, the textual annotation "The figure was created in BioRender" appears superfluous and may be omitted.

Minor changes in style could still improve the paper.

Author Response

Dear reviewer, we appreciate your effort to help us to improve our manuscript.

Figures could benefit from discrete categorization, denoted as A and B.

R: The new version of figures has added A and B.

Within Figure 3, the textual annotation "The figure was created in BioRender" appears superfluous and may be omitted.

R: Thank you for the suggestion, the annotation was deleted.

Reviewer 3 Report

The authors have answered my concerns.

Author Response

The authors have answered my concerns.

R: Dear reviewer, we appreciate your suggestions, really all of them were helpful to improve our manuscript.